# IL-17 Inhibition: A Valid Therapeutic Strategy in the Management of Hidradenitis Suppurativa

**DOI:** 10.3390/pharmaceutics15102450

**Published:** 2023-10-11

**Authors:** Dalma Malvaso, Laura Calabrese, Andrea Chiricozzi, Flaminia Antonelli, Giulia Coscarella, Pietro Rubegni, Ketty Peris

**Affiliations:** 1Dermatologia, Dipartimento di Medicina e Chirurgia Traslazionale, Università Cattolica del Sacro Cuore, 00168 Rome, Italy; malvasodalma@gmail.com (D.M.); laura.calabrese@unisi.it (L.C.); flamiantonelli@gmail.com (F.A.); giuliacoscarella@gmail.com (G.C.); ketty.peris@unicatt.it (K.P.); 2UOC di Dermatologia, Dipartimento di Scienze Mediche e Chirurgiche, Fondazione Policlinico Universitario A. Gemelli-IRCCS, 00168 Rome, Italy; 3Dermatology Unit, Department of Medical, Surgical and Neurological Sciences, University of Siena, 53100 Siena, Italy; pietro.rubegni@gmail.com

**Keywords:** hidradenitis suppurativa, IL-17 inhibitors, biologics, secukinumab, bimekizumab, brodalumab, ixekizumab, CJM112, izokibep, sonelokimab

## Abstract

Hidradenitis suppurativa (HS) is a chronic inflammatory skin disease with a significant negative impact on the quality of life of patients. To date, the therapeutic landscape for the management of the disease has been extremely limited, resulting in a profound unmet need. Indeed, adalimumab, an anti-tumor necrosis factor (TNF)-α monoclonal antibody, is the only approved biologic agent for HS, obtaining a therapeutic response in only 50% of HS patients. Numerous clinical trials are currently ongoing to test novel therapeutic targets in HS. The IL-17-mediated cascade is the target of several biologic agents that have shown efficacy and safety in treating moderate-to-severe HS. Both bimekizumab and secukinumab, targeting IL-17 in different manners, have successfully completed phase III trials with promising results; the latter has recently been approved by EMA for the treatment of HS. The aim of this review is to summarize the current state of knowledge concerning the relevant role of IL-17 in HS pathogenesis, highlighting the key clinical evidence of anti-IL-17 agents in the treatment of this disease.

## 1. Introduction

Hidradenitis suppurativa (HS, also known as acne inversa) is a chronic inflammatory skin disorder affecting approximately 1% of the general population [1]. The peak of prevalence occurs between 20 and 40 years of age, with a decrease after 50 years of age [2].

The disease is clinically characterized by recurrent episodes of neutrophilic inflammation, mostly involving pilosebaceous/apocrine units bearing skin (predominantly axillary and inguinal folds and the perianal area). Starting from the hair follicles, inflammation induces the development of painful nodules and abscesses and, at a later stage, pus-discharging tunnels (sinus tracts or fistulas) and extensive scars. Because of its peculiar and sometimes non-reversible clinical manifestations, HS can detrimentally affect the quality of life of patients, influencing social, personal, and emotional life [3].

Nevertheless, the understanding of HS pathogenesis is still partial, and the therapeutic options remain limited, resulting in a critical unmet clinical need. To date, the treatment of HS ranges from topical and intralesional therapies for mild disease, including topical antibiotics, to systemic drugs, such as various antibiotic regimens, hormonal therapies, retinoids, and immunosuppressive and biologic agents, for moderate-to-severe disease [4]. Both medical and surgical approaches are combined where appropriate. At present, adalimumab, an anti-tumor necrosis factor (TNF)-α monoclonal antibody (mAb), is the only approved biologic agent for HS, achieving a successful response in about half of treated patients [5]. The knowledge of HS pathogenesis is rapidly expanding and, as a result, the treatment of HS is changing, with an emerging emphasis on targeted molecules as the key future treatment approach.

## 2. Current Pathogenetic Model in HS

The pathogenesis of HS has not been completely elucidated, although it is known that a complex interplay between genetic, hormonal, immunological, and microbial factors, together with tobacco smoking and obesity, contributes to disease occurrence and severity [6].

At the mechanistic level, the main events leading to the development of HS encompass (i) an altered infundibular keratinization with consequent hyperkeratosis/occlusion, and (ii) the aberrant activation of innate immune pathways with a massive neutrophil-rich inflammatory infiltrate [7].

Initial events in HS are considered to be mediated by PAMPs (pathogen-associated molecular patterns) and DAMPs (danger-associated molecular patterns) released upon early follicular occlusion and bacterial proliferation, resulting in the inflammasome-mediated release of Interleukin (IL)-1β predominantly by tissue macrophages and the further downstream cytokine release that includes IL-17 and TNFα [8].

In advanced pathogenetic stages, CD4-positive T cells that produce both IL-17 and interferon (IFN)γ, indicative of a Th1/17 phenotype, are enriched in lesions together with high levels of the IL-1 family cytokines IL-1β and IL36, as well as TNFα [6].

Comprehensively, HS exhibits pathogenetic features of both neutrophilic dermatoses (massive neutrophil infiltrations especially in the later stages of the disease) and autoinflammatory disorders (inflammasome-driven IL-1β dominance), with a strong contribution of Th1 and Th17 immune cells [9,10].

Interestingly, HS, far from being a disease limited to the skin, has a well-documented association with a systemic state of inflammation and with metabolic syndrome (MetS), identified even on the molecular level [11]. For example, IL-17 signaling, hyperactive in both HS and MetS, could represent one molecular link between HS and metabolic syndrome [11].

## 3. The Role of IL-17 in the Pathophysiology of HS

During recent years, IL-17 cytokines have emerged as key players in multiple inflammatory disorders, including cutaneous inflammation, thus appearing as an intriguing therapeutic target in dermatology [12].

The IL-17 family comprises multiple members, namely IL-17A, IL-17C, IL-17E, and IL-17F [13]. IL-17A and IL-17F are often co-expressed and secreted predominantly by a subset of CD4+ T helper cells, named Th17 cells, but also, although to a lesser extent, by other cell types such as CD8+ T cells (Tc17) [14]. IL-17C is expressed by epithelial cells, including keratinocytes, in response to Toll-like receptors (TLRs) and cytokine activation [15,16]. 

Furthermore, IL-17E, also known as IL-25, can be produced by keratinocytes and endothelial cells and plays a prominent role in inducing Th2 immune responses [17].

TGFβ, IL-6, IL-21, and IL-23 induce the differentiation and expansion of Th17 cells [18]. These cells produce IL-17A, which in turn can stimulate the production of a number of chemokines (such as CCL20 and neutrophil-attracting chemokines that include CXCL1 and CXCL8), cytokines (such as G-CSF and IL-19), and epidermal antimicrobial proteins (AMPs) [19,20]. Therefore, IL-17 cytokines exert a multitude of pro-inflammatory effects, including the chemotaxis of monocytes and neutrophils in the skin, the recruitment of Th17 and myeloid cells, and further increases in IL-17 production and immune cell infiltrations in HS lesions, in a feed-forward inflammatory loop [21].

Interestingly, IL-17 has been shown to activate keratinocyte proliferation in an indirect manner by inducing the expression of IL-19 and heparin-binding EGF-like growth factor (HBEGF) in these cells, thus stimulating epidermal hyperplasia [22,23,24], which is a common histopathological finding in HS lesions [25]. Furthermore, IL-17 is able to promote various inflammatory processes in the epidermis, such as the expression of IL-1β [26].

Recently, increasing scientific evidence has emerged to endorse the role of IL-17 in the pathogenesis of this disease.

In 2011, Wolk et al. demonstrated an upregulation of the mRNA of IL-17A in the lesions of patients with HS and psoriasis, while the levels in control subjects and atopic dermatitis patients were not increased [27]. 

Schlapbach et al. investigated the role of the IL-23/Th17 pathway in HS utilizing semiquantitative real-time polymerase chain reaction (RT-PCR) and immunohistochemistry (IHC), reporting a high expression of both IL-23 and IL-17 in HS lesions. On the other hand, the immunohistochemical analysis of HS lesional and healthy skin showed a marked infiltration by IL-17-producing cells in lesional skin, confirming the PCR results [28]. Similarly, other studies have further suggested an enhanced expression of IL-17A and IL-17F mRNA levels in HS lesional skin in comparison with healthy controls [21,29].

Moreover, Lima et al. performed IHC staining and Western blot experiments to investigate the localization and expression of IL-17 in HS skin and detected a number of IL-17+ cells that were significantly increased in lesional and perilesional skin, compared to healthy controls; interestingly, neutrophils were identified as a major source of IL-17 in HS skin [30]. 

Another study performed IHC and mRNA analyses on lesional HS skin and detected increased mRNA levels of IL-17A, IL-17C, and IL-17F in comparison with controls, as well as an in situ localization of IL-17C in the supra-basal epidermis, with particular accentuation of the stratum corneum and stratum granulosum, as well as in the dermis [31]. 

Increased circulating IL-17 levels have been identified in patients with HS, with higher serum concentrations detected in patients with a more advanced disease [32].

An in-depth analysis of HS serum proteome based on proximity extension assay technology found an upregulation of IL-17A and IL-17D levels compared to healthy controls. Moreover, the serum levels of these cytokines were found to be more elevated in HS samples with histologically confirmed tunnels compared to non-tunnel samples and in draining versus non-draining tunnel samples [33].

In another study, the proteomic profile of HS was investigated in frozen skin samples using Olink high-throughput technology, showing an increased protein expression of IL-17A in HS compared with healthy controls, which was positively correlated with disease severity measured using the International Hidradenitis Suppurativa Severity Score System (IHS4) [34,35]. 

A recent study on skin samples from 40 patients with HS, 4 patients with psoriasis, and 29 healthy controls identified IL-17 signaling as being a potential strong contributor to HS based on transcriptomic data analysis. Interestingly, the predominant IL-17A+ cell population in HS was identified as mast cells [36].

Furthermore, another study analyzed single-cell transcriptomic data from HS skin obtained during deroofing dermal tunnels and compared them to psoriasis and healthy skin. Interestingly, in the HS samples, there was a predominance of the IL-1β–T17 cell cytokine axis, indeed suggesting that T17 cells are likely to be activated by IL-1β in HS, while psoriasis T17 cells are activated by IL-23 signaling [37]. These results prompt the consideration that agents targeting IL-1β and IL-17 should be more successful than those that target IL-23 for the treatment of HS [38].

Similar results were obtained in another single-cell transcriptomic study, which demonstrated that cells that express either IL-17A or IL-17F are expanded even in early HS lesions, and IL-17F-producing cells can co-localize with IL-1-expressing cells in chronic lesions. These observations regarding early and chronic HS lesions support the important role of both IL-17A and IL-17F in the pathogenesis of HS and further corroborate the hypothesis that elevated IL-1 may partly explain the high expression of IL-17F in HS lesions [39].

As mentioned above, the well-documented association between HS and a systemic state of inflammation, or even MetS, could possibly have a molecular basis. Interestingly, cytokines such as TNFα, IL-1β, IL-6, IL-8, and IL-17A, which are elevated in MetS and cardiovascular diseases, are also overexpressed in HS [11,40]. However, it remains unclear whether the presence of such elevated inflammatory markers in HS is a causal factor that triggers MetS or whether both conditions share a similar pathophysiology.

Taken together, all of this evidence sheds light on IL-17 as a key player in the pathogenesis of HS (Figure 1), alongside other inflammatory skin diseases, and has paved the way for the investigation of IL-17-targeted agents in a clinical trial setting.

## 4. Materials and Methods

The aim of this narrative review was to describe the pathogenetic role of IL-17 in HS and to examine novel therapeutic strategies that are currently in clinical development for the treatment of HS, with a focus on IL-17 inhibitors.

We searched the published literature for relevant original articles, case series, and real-world studies using databases, namely PubMed and Cochrane library, up to 9 August 2023. We also searched clinicaltrials.gov for the term ‘hidradenitis suppurativa’ on 9 August 2023, including drugs with ongoing or recently completed trials; trials that had been withdrawn or whose status was unknown were excluded. 

## 5. IL-17-Targeting Agents Currently in the Pipeline for the Treatment of HS

The deeper understanding about the role of pivotal pro-inflammatory cytokines involved in the multifaceted pathogenesis of HS led to the potential efficacy of novel biologics and other targeted therapies in the treatment of moderate-to-severe HS being evaluated. In particular, numerous clinical trials are currently ongoing to assess the potential clinical benefit of IL-17 inhibitors in the HS treatment landscape (Table 1 and Figure 2).

### 5.1. Secukinumab

Secukinumab, a human IgG1/κ monoclonal antibody that selectively binds to IL-17A, was investigated in two randomized placebo-controlled phase III studies (SUNSHINE, SUNRISE) [41,42]. Both trials evaluated the short- (16 weeks) and long-term (up to 52 weeks) efficacy, safety, and tolerability of secukinumab in 541 (SUNSHINE) and 543 (SUNRISE) subjects, respectively. Following a loading dose of five weekly injections of secukinumab 300 mg subcutaneously (SC), participants in the active arms were randomized to receive either secukinumab 300 mg SC every four weeks (Q4W) or every 2 weeks (Q2W) until week 16. The primary endpoint was the proportion of patients achieving an HS clinical response (HiSCR) after 16 weeks. A higher proportion of patients in both active arms achieved HiSCR by week 16 compared with placebo, although only the Q2W dosing regimen achieved a significant difference vs. placebo (SUNSHINE trial: 45% Q2W secukinumab vs. 34% placebo, *p* = 0.007; 42% Q4W secukinumab, *p* = 0.042), while statistical significance was observed with both treatment regimens in SUNRISE (42% Q2W secukinumab, *p* = 0.015; 46% Q4W secukinumab, *p* = 0.002; vs. 31% placebo) [43]. HiSCR response was sustained up to week 52 in SUNSHINE and SUNRISE with Q2W and Q4W secukinumab compared with placebo (76% and 84% with Q2W secukinumab, respectively; 81% and 77% with Q4W secukinumab, respectively). Overall, Q2W met the primary endpoint in both clinical studies. Of note, efficacy demonstrated by both secukinumab dosing regimens by week 16 was maintained until week 52, showing sustained improvement beyond week 16. Secondary endpoints included percentage change from baseline in abscess and inflammatory nodule (AN) count and participants experiencing HS flares over 16 weeks. In both SUNSHINE and SUNRISE, Q2W secukinumab was superior to placebo in terms of the percentage change from baseline in AN count by week 16. In the SUNSHINE study, significantly less participants in the Q2W secukinumab group than in the placebo arm experienced a flare during the first 16 weeks. On the contrary, in the SUNRISE study, there was no significant difference between the two groups. Safety data from these trials were consistent with the secukinumab safety profile across other indications and previously published data. The most common adverse events reported in both trials were headache, nasopharyngitis, and HS worsening. Efficacy in terms of HiSCR maintenance was sustained up to week 52 in SUNSHINE and SUNRISE with Q2W and Q4W secukinumab being compared (76% and 84% with Q2W secukinumab, respectively; 81% and 77% with Q4W secukinumab, respectively). Secondary endpoints included percentage change from baseline in abscess and inflammatory nodule (AN) count and participants with HS flares. Secukinumab Q2W was superior to placebo in both the SUNSHINE (−46.8% in the active arm vs. −24.3% in the placebo group; *p* < 0.001) and SUNRISE (−39.3% in the active arm vs. −24.3% in the placebo group; *p* < 0.0001) studies in terms of the percentage change from baseline in AN count by week 16. Secukinumab Q4W did not significantly improve AN by week 16 compared to placebo in SUNSHINE (−42.4% in the active arm vs. −24.3% in the placebo arm; *p* = 0.0004). In SUNRISE, there was a significant difference between the two cohorts (−45.5% in the secukinumab Q4W arm vs. −22.4% in the placebo arm; *p* = 0.0001). In the SUNSHINE study, significantly fewer patients in the secukinumab Q2W group experienced a flare during the first 16 weeks compared to the placebo group. Safety data from these trials were consistent with secukinumab safety profiles across other indications and previously published data. 

An extension study was designed to assess the maintenance of HiSCR response by week 104 with either continuous or discontinued therapy with secukinumab in participants who completed either of the two phase III trials [44]. Patients who achieved HiSCR after 52 weeks of treatment were randomized in a 2:1 ratio to receive for a further 52 weeks either secukinumab at one of the two doses assigned to the core trials or placebo, to evaluate the loss of response (LOR). Based on the strength of the consistent results from the phase III program, secukinumab has been approved by the European Commission for the treatment of adults with active moderate-to-severe HS unresponsive to conventional systemic therapy [45].

Several real-world studies have been published demonstrating the efficacy of secukinumab in the management of HS. Among these, an Italian retrospective multicenter study assessed the efficacy of secukinumab 300 mg in 31 patients with HS who had failed or had contraindications to at least one TNF-alfa treatment. The primary endpoints were the achievement of HiSCR by week 28, the reduction of the total number of abscesses and inflammatory nodules (AN), and the improvements in Hidradenitis Suppurativa Severity Score System (IHS4) and Dermatology Life Quality Index (DLQI) scores. By week 28, 41% of participants had achieved HiSCR. In parallel, a decrease in IHS4 and AN count was recorded throughout the trial. The drug resulted in a positive effect of improving the quality of life of affected patients, as demonstrated by the reduction in DLQI from 17 (range 4–30) at baseline to 12 (range 5–15) by week 28 [46].

### 5.2. Bimekizumab 

Bimekizumab is a humanized IgG1 antibody, neutralizing both IL-17A and IL-17F, and has been shown to be effective in the management of psoriasis and psoriatic arthritis [47,48].

The results of a randomized phase II trial involving 90 patients with moderate-to-severe HS demonstrated clinically meaningful improvements obtained with bimekizumab [49,50]. Participants were randomly assigned to bimekizumab (640 mg by week 0, 320 mg Q2W), placebo, or adalimumab in a 2:1:1 ratio. The percentage of patients achieving HiSCR by week 12 was the main efficacy endpoint. Additional efficacy outcomes included HiSCR75 and HiSCR90, defined as a ≥75% or ≥90% reduction, respectively, in total abscess and inflammatory nodule (AN) count with no increase in abscess or draining tunnel count relative to baseline and the International Hidradenitis Suppurativa Severity Score (IHS4). The bimekizumab-treated group had a higher success rate (57.3%) than the placebo group (26.1%). Similarly, a greater proportion of participants receiving bimekizumab achieved HiSCR75 and HiSCR90 by week 12 (46% and 32%, respectively, compared to 10% and 0% in the placebo group and 35% and 15% in the adalimumab arm). Comparable rates of TEAEs were seen in all treatment groups and most were classified as mild or moderate. 

Following the promising results of phase II, a phase III clinical trial program (BE HEARD I, BE HEARD II, BE HEARD EXT) is currently ongoing to investigate the efficacy and safety of bimekizumab in the treatment of moderate-to-severe HS. The preliminary published data have demonstrated the efficacy of bimekizumab in achieving the primary endpoint of HiSCR by week 16 compared to placebo in both HEARD I (*n* = 505) and HEARD II (*n* = 509) (47.8% vs. 28.7% and 52% vs. 32.2%, respectively) [51,52]. Two bimekizumab dosing regimens (320 mg Q2 and 320 mg Q4) were compared with placebo during the 16-week initial and 32-week maintenance treatment periods in both BE HEARD I and II. A greater proportion of subjects receiving monthly bimekizumab compared to placebo achieved HiSCR50 by week 16, with statistical significance only attained in BE HEARD II (53.8% vs. 32.2%). BE HEARD II met the key secondary endpoint of HiSCR75 by week 16 for both treatment arms versus placebo, whereas BE HEARD I only met the endpoint for bimekizumab 320 mg Q2 compared to placebo, with sustained responses up to week 48.

Overall, these results demonstrate that treatment with bimekizumab provided a significant improvement in HS-related clinical manifestations and health-related quality of life by week 16 compared with placebo, with sustained efficacy until week 48.

The drug was generally well tolerated, with HS worsening, headache, oral candidiasis, and diarrhea being the most common adverse events reported. The safety profile was consistent with previously reported data.

### 5.3. Brodalumab

Brodalumab is a fully human IgG2 monoclonal antibody that targets the IL-17 receptor A (IL-17RA) subunit, thereby blocking the signaling of multiple isoforms of IL-17 (IL-17A, IL-17F, IL-17C, and IL-17 A/F). It is currently approved for the treatment of moderate-to-severe plaque psoriasis. A pilot open-label cohort study was designed to assess safety and identify biomarkers of disease activity in brodalumab-treated subjects [53]. Ten patients with no history of inflammatory bowel disease (IBD) received 210 mg brodalumab every two weeks for 24 weeks. The primary endpoints were the evaluation of changes in the lesional tissue levels of IL-17A, IL-17C, IL-17F, and IL-23 by weeks 12 and 24 by performing skin biopsies at baseline and weeks 4 and 12. RNA sequencing and immunohistochemistry of non-lesional, perilesional, and lesional HS skin biopsies, as well as Olink high-throughput proteomics of serum at baseline and weeks 4 and 12, were performed. The histological findings were consistent with a reduction in psoriasiform epidermal hyperplasia observed at baseline and a decrease in dermal inflammatory infiltrates after 12 weeks of brodalumab therapy [54]. Skin biopsies taken in week 12 showed that patients who had higher baseline expression of neutrophil-associated lipocalin 2 (LCN2) in the skin or IL-17A in the serum had greater reductions in HS-related inflammatory cytokines [55].

An improvement in clinical disease activity was assessed by the number of patients achieving HiSCR by weeks 12 and 24 compared to baseline, as well as the Sartorius score and IHS4. Clinical response was rapid and substantial, with all participants (100%) achieving HiSCR by week 12 and week 24, and a 75% reduction in abscess and nodule count (AN) and a 100% reduction in AN being seen in 60% and 40% of participants, respectively, by these time points. No serious TEAEs, suicidal ideation, or depression were observed in the study cohort.

In another small pilot study, participants received weekly doses of brodalumab [56]. Blood and tissue samples were collected to better define the molecular mechanisms underlying treatment response, to identify the biomarkers that are correlated with disease severity and to better understand HS pathogenesis. No results are currently available.

Real-world case reports suggest the clinical benefit of brodalumab in the challenging management of HS. Yoshida et al. described the remarkable outcome of one patient with long-standing refractory HS and psoriasis, both successfully treated with brodalumab [57]. Similarly, Arenbergerova at al. report on one case of severe extensive gluteal HS treated with brodalumab after the failure of anti-TNF-alpha therapy [58]. The patient experienced a marked improvement in clinical manifestations, including a reduction in inflammatory lesions, as evidenced by a decrease in IHS4 score from 62 to 18, together with a reduction in laboratory findings of systemic inflammation.

### 5.4. CJM112

CJM112 is a fully human anti-IL-17A IgG1/κ monoclonal antibody which binds to both IL-17A and IL-17AF with a similar degree of affinity. Its safety and efficacy were assessed in a 16-week, double-blind, placebo-controlled phase II trial, followed by an extension period of 16 weeks [59]. CMJ112 300 mg was administered subcutaneously on a weekly basis for the first 5 weeks, followed by injections every 2 weeks (Q2W) up to week 16. Sixty-six participants, aged 18–65 years, with moderate-to-severe HS were recruited. The primary outcome was to determine the efficacy of CJM112 300 mg as measured using the HS-PGA responder rate (≥2 point decrease in HS-PGA score by week 16). By week 16, there were significantly more HS-PGA responders in the CJM112 300 mg treatment arm than in the placebo group (32.3% vs. 12.5% (4/32); *p* = 0.03) [60]. However, a higher-than-expected placebo effect was reported after 16 weeks of placebo treatment, with a decrease in the mean baseline HS-PGA score. There were no significant differences in HS-PGA achievement between the CJM112 300 mg and the placebo groups by the other time points (weeks 2, 4, and 12). Overall, CJM112 was well tolerated, and its safety profile was comparable to that of a placebo. Nasopharyngitis (16.7%), nausea (12.1%), diarrhea (10.6%), and headache (10.6%) were the most frequent AEs. In comparison to the placebo group, the incidence of nasopharyngitis and nausea was greater in the CJM112 300 mg group.

### 5.5. Izokibep

Izokibep is engineered using Affibody^®^ molecules, comprising small, triple-helical protein domains, binding to IL-17A with high affinity due to its small molecular size, acting as a selective and potent inhibitor [61]. Izokibep combines two IL-17A-specific Affibody domains with an albumin-binding domain (ABD) (Albumod^®^; Affibody Medical AB, Solna, Sweden), creating an IL-17A ligand trap. The drug acts as a novel subcutaneous (SC) IL-17A inhibitor with a small molecular size which enables excellent biodistribution to inflammatory sites, while its reduced molecular size facilitates high drug exposure through SC injection.

The drug is being evaluated in a randomized, double-blind phase IIb study in moderate-to-severe HS. The study has enrolled 180 patients and is evaluating izokibep versus placebo in achieving HiSCR75 by weeks 12 and 16 [62]. Early results observed in week 12 revealed that 71% of the participants achieved HiSCR50, 57% achieved HiSCR75, 38% achieved HiSCR90, and 33% achieved HiSCR100 [63]. The drug was well tolerated, with injection site reactions being the most reported adverse event. No Candida events were registered up to week 12. Two serious adverse events were observed, including inflammatory bowel disease in one patient with previous symptoms, which was evaluated as being possibly related to the study drug. Another participant with previous symptoms and known diverticulosis developed a peri-colonic abscess, which was reported as being unrelated.

The double-blind, placebo-controlled phase III trial is currently recruiting participants with moderate-to-severe HS [64] to evaluate the percentage of subjects achieving HiSCR75 by week 16. Eligible participants are TNFα inhibitor (TNFi)-naive, with previous inadequate response or intolerance to TNFi, or those contraindicated for TNFi.

### 5.6. Sonelokimab

Sonelokimab is a novel trivalent nanobody consisting of domains specific for IL-17A, IL-17F, and human serum albumin, with the latter allowing for an increased concentration of the drug at the sites of inflammatory oedema. Nanobodies are a novel class of therapeutic proteins based on single-domain antibodies that contain only the heavy chain.

A double-blind, placebo-controlled phase II trial assessed the clinical efficacy and safety of sonelokimab across two dose regimens (120 mg and 240 mg s.c.) compared with placebo and adalimumab in 234 participants with moderate-to-severe HS [65]. The primary outcome of the trial was evaluating the percentage of subjects achieving HiSCR75 by week 12. Recently released preliminary data revealed that a significantly greater proportion of patients treated with both sonelokimab 120 mg and 240 mg achieved HiSCR75 by week 12 compared to placebo, meeting the primary endpoint [66]. By week 12, other secondary endpoints, such as HiSCR90 and his4, also attained statistical significance, with clinically considerable improvements. Sonelokimab demonstrated a favorable safety profile consistent with the well-established profile of IL-17 antagonists, with no new safety signals reported.

Treatment with subcutaneous sonelokimab is currently being investigated in plaque psoriasis and has demonstrated a significant benefit over placebo, with rapid and sustained clinical improvements and an acceptable safety profile [67].

## 6. Conclusions

HS is characterized by complex genetic, molecular, and phenotypic heterogeneity, which is reflected in a variable response to therapy. A broader understanding of the pathophysiology of HS will allow treatment to be increasingly tailored to individual endotypes. Therefore, the identification of predictive molecular biomarkers of drug response, including anti-TNF or anti-IL-17 biologics, is of great importance. With adalimumab achieving a successful response in approximately 50% of HS patients, further therapeutic options are warranted, and IL-17 inhibitors offer an emerging frontier in HS management.

Comparative studies of both anti-TNF and anti-IL-17 molecules are currently limited. A recent network meta-analysis assessing the efficacy and safety of various biologic antibodies and other drugs under investigation for HS provided some notable findings. Notably, the efficacy of bimekizumab, as assessed by the HiSCR in weeks 12–16, was statistically non-inferior to that of adalimumab [68].

In addition, there was no significant difference between adalimumab and bimekizumab in the probability of achieving DLQI 0/1 by weeks 12 to 16. Addressing the safety profiles, adalimumab, bimekizumab, and secukinumab were found to be comparable to the placebo, suggesting that they are generally well tolerated by patients [69]. 

An additional meta-analysis supported these findings, showing that both adalimumab and bimekizumab were superior to the placebo in achieving HiSCR by weeks 12 to 16, with no significant difference observed between the two drugs. Interestingly, in a multicenter phase II study designed to evaluate the effects of bimekizumab in the context of HS, the rates of treatment-emergent adverse events (TEAEs) appeared to be similar in the bimekizumab, adalimumab, and placebo groups. However, it is important to note that Candida infections were observed in the bimekizumab group, as expected for IL-17 inhibitors [50].

The abovementioned findings contribute to our understanding of the comparative efficacy and safety of anti-TNF and anti-IL-17 therapies in the treatment of HS and highlight the potential of the latter as a promising alternative.

The clinical outcomes mostly deriving from a trial setting strengthen the role of IL-17 in the pathogenesis of HS and highlight the potential of targeting this cytokine to offer patients a novel, effective, and well-tolerated treatment option in the management of this challenging disease. However, patients with a very severe condition, showing a high number of inflammatory lesions, have been excluded from clinical trials and this subset of patients could potentially benefit from an IL-17 blockade, as the IL-17 signal is suggested to be enhanced in skin samples with advanced Hurley stages [70]. Moreover, to our knowledge, no studies have been conducted to compare clinical responses to anti-IL-17 drugs across different HS phenotypes. Therefore, further studies are needed to better define the patient profile of the best responders to IL-17 inhibition. 

The pathogenesis of HS is influenced by multifaceted interplay among genetic, immunological, and microbial factors, as well as smoking.

Genetic defects in γ-secretase subunits can result in abscess formation. Smoking can activate keratinocytes, leading to hyperkeratosis and follicular occlusion. Changes in the skin microbiome lead to the overexpression of PAMPs (pathogen-associated molecular pattern molecules).

In hidradenitis suppurativa, immunomodulatory functions include the activation of various immune cells, including macrophages, neutrophils, T cells, and dendritic cells (DC), and the production of cytokines. A critical role is played by IL-17A, which is mainly produced by neutrophils, mast cells, and Th17 cells. IL-17A amplifies the inflammatory process by stimulating neutrophils and macrophages and promoting the production of PAMPs. These events can ultimately lead to the activation of keratinocytes and the development of hyperkeratosis, which contribute to a feed-forward mechanism in hidradenitis suppurativa.

## Figures and Tables

**Figure 1 pharmaceutics-15-02450-f001:**
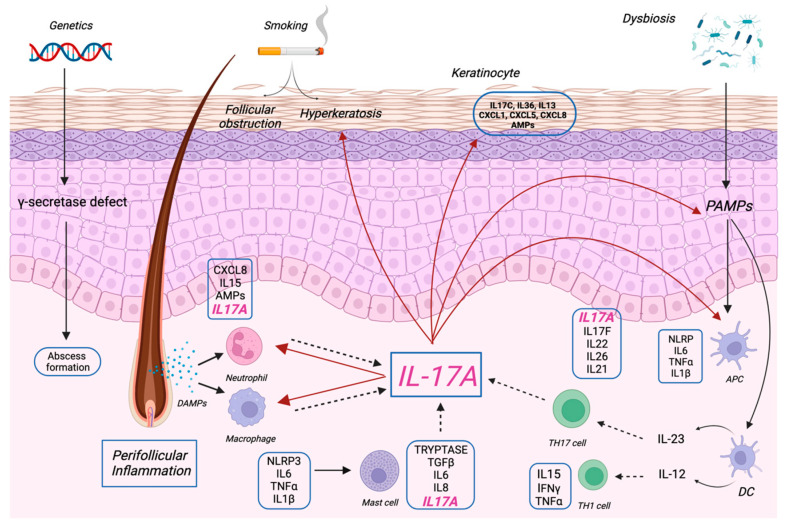
Pathogenic role of IL-17A in hidradenitis suppurativa.

**Figure 2 pharmaceutics-15-02450-f002:**
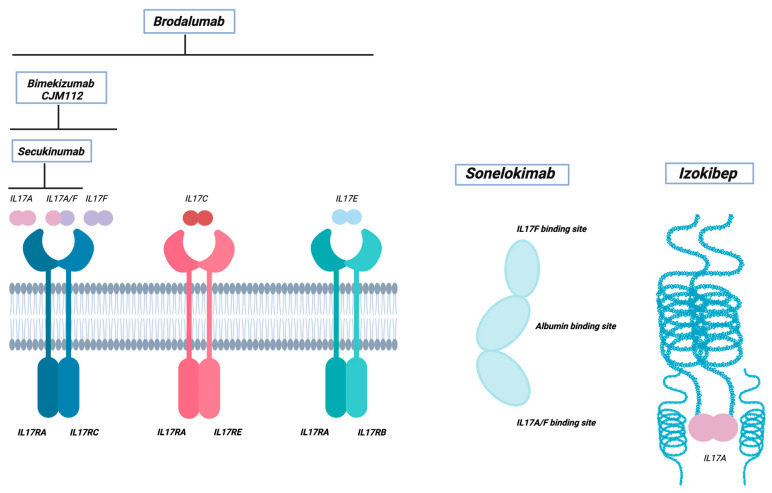
Specific therapeutic agents in development for the treatment of hidradenitis suppurativa. Brodalumab, a fully human monoclonal antibody that specifically targets the Interleukin 17 Receptor A (IL17RA), inhibits the downstream signaling of multiple IL-17 family cytokines. Bimekizumab and CJM112 are humanized antibodies that inhibit the signaling of both IL-17A and IL-17F. Secukinumab is a fully human monoclonal antibody that selectively binds to IL17A homodimers or heterodimers. Sonelokimab is a trivalent nanobody composed of IL-17A, IL-17F, and albumin-binding domains, effectively blocking the IL-17A and IL-17F downstream pathways. Izokibep owns a triple-helical structure, enabling it to bind IL17A with high affinity, thereby functioning as a neutralizing IL-17A ligand. Abbreviations: IL, Interleukin.

**Table 1 pharmaceutics-15-02450-t001:** Emerging IL-17-targeting agents for HS.

Drug Name	MoA	Study	Clinical Trial No.	Status	Primary Endpoint (s)
**Secukinumab**	IL-17A inhibitor, mAb	Phase III, randomized, double-blind, placebo-controlled (SUNRISE)	NCT03713632	Completed	Proportion of participants with HiSCR by week 16
		Phase III, randomized, double-blind, placebo-controlled (SUNSHINE)	NCT03713619	Completed	Proportion of participants with HiSCR by week 16
		Phase III, double-blind, randomized withdrawal extension study	NCT04179175	Active; not recruiting	Time to loss of response (LOR) in HiSCR responders (weeks 52–104)
**Bimekizumab**	IL-17 A/F inhibitor, mAb	Phase III, open-label, parallel group, extension (BE HEARD EXT)	NCT04901195	Active; not recruiting	Percentage of participants with TEAEs during the study
		Phase III, randomized, double-blind, placebo-controlled (BE HEARD II)	NCT04242498	Completed	Percentage of participants with HiSCR50 by week 16
		Phase III, randomized, double-blind, placebo-controlled (BE HEARD I)	NCT04242446	Completed	Percentage of participants with HiSCR50 by week 16
**Brodalumab**	IL-17 inhibitor, mAb	Early phase I	NCT03960268	Completed	Biomarkers by weeks 12 and 24; incidence of TEAEs
		Early phase I	NCT04979520	Completed	IL-17RA saturation during brodalumab administration by week 12 vs. baseline
**CJM112**	IL-17A inhibitor, mAb	Phase II, randomized, double-blind, placebo-controlled	NCT02421172	Completed	Clinical response rate by week 16
**Izokibep**	IL-17A inhibitor, fusion protein	Phase IIb, randomized, double-blind	NCT05355805	Active; not recruiting	Part A: percentage of subjects achieving HiSCR75 by week 12. Part B: percentage of subjects achieving HiSCR75 by week 16
		Phase III, randomized, double-blind, placebo-controlled	NCT05905783	Recruiting	Percentage of subjects achieving HiSCR75 by week 16
**Sonelokimab**	IL-17 A/F inhibitor, mAb	Phase II, randomized, double-blind, placebo-controlled	NCT05322473	Active; not recruiting	Percentage of subjects achieving HiSCR75 by week 12

HiSCR, Hidradenitis Suppurativa Clinical Response; IL, Interleukin; IL-17RA, Interleukin-17 receptor A; mAb, monoclonal antibody; MoA, mechanism of action; TEAEs, treatment-emergent adverse events.

## Data Availability

Not applicable.

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
