# Peer review of "IL-17 Inhibition: A Valid Therapeutic Strategy in the Management of Hidradenitis Suppurativa"

_pharmaceutics, 2023, doi:10.3390/pharmaceutics15102450_

Round 1

Reviewer 1 Report

Interesting review on an intriguing therapeutic target, IL17 or its receptor. My personal experience with the drug in HS does not deviate much from the results obtained with adalimumab. In fact, no clinical trial has been compared with this one, only with a placebo. The safety profile could be interesting and differentiating with the anti-IL17s, something especially noteworthy. However, we still do not know the phenotype of HS that responds best to anti-IL17s; perhaps this could have been mentioned in the conclusions, given that there is no discussion in the manuscript due to the article type.

Author Response

Response: Thank you very much for your comment. A short paragraph was added to analyze this aspect.

Reviewer 2 Report

I am very glad to review this article. It is well-known that patients suffering from hidradenitis suppurativa (HS) demonstrate a molecular profile in keeping with a state of systemic inflammation and are often found to fit the criteria for a diagnosis of metabolic syndrome (MetS). So, I think that it is better to incorporate the bridge role of IL-17 in both HS and MetS in the two sections: “Current pathogenetic model in HS” and “The role of IL-17 in the pathophysiology of HS”.

The reference styles should be consistent. For example, upper and lower case in the titles of  some references.

Author Response

Reviewer 2.

I am very glad to review this article. It is well-known that patients suffering from hidradenitis suppurativa (HS) demonstrate a molecular profile in keeping with a state of systemic inflammation and are often found to fit the criteria for a diagnosis of metabolic syndrome (MetS). So, I think that it is better to incorporate the bridge role of IL-17 in both HS and MetS in the two sections: “Current pathogenetic model in HS” and “The role of IL-17 in the pathophysiology of HS”.

Response: Dear reviewer, thank you very much for your comment. This is a very interesting point. I have added a section discussing the possible molecular link between these two diseases in the paragraph “Current pathogenetic model in HS” and “The role of IL-17 in the pathophysiology of HS”.

The reference styles should be consistent. For example, upper and lower case in the titles of  some references.

Response: The reference styles have been fixed accordingly.

Reviewer 3 Report

The current manuscript provides an overview of a treatment strategy using IL-12 inhibitors for the management pf hidradenitis suppurativa (HS). This is interesting and relevant as well, as it can potentially help clinicians and researchers better understand the current status of biologic therapies for HS. A few small issues in the current format have to be addressed.

1). The current manuscript focuses on IL-17 inhibition. However, the exact the IL-17 signaling pathway (e.g., IL-17 receptor) is not fully covered. It is better to be cautious when “pathway” is used, as targeting IL-17 is the major topic in this work.

2). A brief discussion of current options for treatment of HS is necessary in Introduction, as this will be extremely helpful to illustrate the importance of IL-17 targeting for HS therapy.  

3). In Table 1, most inhibitors are actually monoclonal antibodies, which should be added to MoA. The term “small molecule” for Izokibep may not be appropriate, as this typically refers to chemical compound inhibitors or ones WM less than 1, 000 Dalton. The reference numbers need to be added to Status.

4). Some minor language issues, for example line105-106, have to be fixed.

The quality of English language is OK, with only minor corrections being necessary. 

Author Response

The current manuscript provides an overview of a treatment strategy using IL-12 inhibitors for the management pf hidradenitis suppurativa (HS). This is interesting and relevant as well, as it can potentially help clinicians and researchers better understand the current status of biologic therapies for HS. A few small issues in the current format have to be addressed.

1). The current manuscript focuses on IL-17 inhibition. However, the exact the IL-17 signaling pathway (e.g., IL-17 receptor) is not fully covered. It is better to be cautious when “pathway” is used, as targeting IL-17 is the major topic in this work.

Response: Dear reviewer, thank you for your comment. The word “pathway” has been better defined in the text.

2). A brief discussion of current options for treatment of HS is necessary in Introduction, as this will be extremely helpful to illustrate the importance of IL-17 targeting for HS therapy.  

Response: Dear reviewer, thank you for your comment. A brief paragraph was added in the introduction regarding the current treatment guidelines in the management of HS and the unmet needs in therapy.

3). In Table 1, most inhibitors are actually monoclonal antibodies, which should be added to MoA. The term “small molecule” for Izokibep may not be appropriate, as this typically refers to chemical compound inhibitors or ones WM less than 1, 000 Dalton. The reference numbers need to be added to Status.

Response: Dear reviewer, thank you for your comment. The mechanism of action of the drugs mentioned was included in the table, and the description of izokibep's molecular structure was expanded in the dedicated section.

4). Some minor language issues, for example line105-106, have to be fixed.

Response: Language issues have been fixed accordingly.